# Research on assessment method of maximum distributed generation hosting capacity in distribution system with high shares of renewables and power electronics

Dai Wan[1], Kailun Fan[1], Jingyu He[2]*, Haochong Zhang[3], Gexing Yang[3], Xujin Duan[1]

**1** State Grid Hunan Electric Power Company Limited Research Institute, Changsha, China, **2** College of Electrical and Information Engineering, Hunan University, Changsha, China, **3** College of Electrical, Electronic and Physical Sciences, Fujian University of Technology, Fuzhou, China

* hjy397766@gmail.com

**Data Availability Statement:** All relevant data are within the manuscript and its Supporting information files.

## Abstract

With the rapid development of distributed generation (DG) within the framework of modern power systems, accurately assessing the maximum DG hosting capacity in distribution networks is crucial for ensuring the safe and stable operation of the power grid. This paper first introduces an assessment model of maximum DG hosting capacity in distribution network based on optimal power flow (OPF). Then, a two-step method that combines the linearization method and the recursive method is proposed, which consists of two parts: firstly, using linearization method to quickly calculate the preliminary assessment value of maximum DG hosting capacity, and then using a recursive method to accurately correct the preliminary assessment value. Additionally, the proposed improved comprehensive sensitivity index and safety constraint verification method can enhance the computational efficiency and accuracy of the recursive algorithm. Finally, the proposed methods were simulated and validated on the IEEE 33-bus system.

## Introduction

Under the goal of 'striving to peak carbon dioxide emissions by 2030 and achieving carbon neutrality by 2060' [1, 2], the power industry plays a pivotal role in carbon emissions and bears the primary responsibility for realizing these objectives. DG has become a crucial focus for the development of the world's future energy and power system [3]. However, due to the intermittent and fluctuating nature of DG power output, its large-scale integration poses significant challenges to the distribution network, threatening the safety and reliability of electricity supply. As the proportion of DG integrated into the distribution network reaches a certain threshold, it substantially impacts planning, design, operation, and maintenance of power system.

Researching assessment methods for the maximum DG hosting capacity in distribution networks can provide data support for the smooth progress of energy transition, promoting the widespread adoption of clean energy. On the one hand, the renovation and expansion of

**Funding:** This work is supported by State Grid Hunan Electric Power Company Science and Technology Project "Technology and Demonstration of Distributed Flexible Resource Cluster Autonomy and Provincial Local-County Collaborative Dispatch in Active Distribution Networks" (5216A522000M).

**Competing interests:** The authors have declared that no competing interests exist.

distribution network are crucial components of the energy transition process and must be planned based on grid load and the requirements for integrating DG. By assessing the maximum DG hosting capacity, rational planning for the renovation and expansion of the power grid can be achieved. On the other hand, accurately assessing the maximum DG hosting capacity in distribution network aids in studying the safety boundaries of distribution network and identifying potential weak points. Therefore, within the context of the widespread integration of DG into distribution network, researching assessment methods for the maximum DG hosting capacity in distribution network holds significant importance.

Based on this, relevant scholars at home and abroad have done a lot of research on distribution network hosting capacity assessment. The literature [4, 5] evaluated the distributed photovoltaic (PV) consumption capacity of distribution networks based on a data-driven approach. The literature [6] considers the distribution network voltage quality, transformer and line transmission capacity constraints, based on stochastic scenario simulation and fixed-order division assessment to achieve the assessment of distribution network PV consumption capacity. The literature [7] adopts an iterative calculation method for the PV hosting analysis by gradually increasing the customer penetration. The literature [8] applied Monte Carlo simulation method to evaluate the new energy consumption capacity of regional distribution networks. Some literature considers employing active network management to enhance the DG hosting capacity of distribution networks. The literature [9] improves the hosting capacity of distribution network for PV by implementing energy storage system on distribution network. The literature [10] establishes a new energy hosting capacity assessment model for distribution networks considering demand side management and network reconfiguration. In addition to considering the fundamental safety constraints of the distribution network, the literature [11] introduced the network loss rate as an indicator and utilized second-order cone relaxation to solve the maximum DG hosting capacity model, in order to meet the operational constraints of the distribution network. The literature [12] proposed a renewable energy consumption capacity assessment method based on system flexibility sufficiency to support renewable energy consumption based on quantitative analysis of system resource power regulation range. At present, most literature adopts linear optimization models, which can improve computational efficiency but may result in certain computational errors. The literature [7] did not consider the operational characteristics of each node. The literature [8] will be affected by the upper limit of customer penetration during calculation, causing the assessment result to be "pseudo" hosting capacity. The literature [11, 12] considers additional optimization objectives, leading to conservative calculation results.

With the continuous development of the distribution network, there is an increasing demand for a real-time and accurate assessment of the maximum DG hosting capacity in the distribution network. In response to this demand, this paper proposes a two-step method for assessing the maximum DG hosting capacity in distribution network based on the above literature. The contributions of this paper are summarized as follows:

- a two-step method for assessing the maximum DG hosting capacity in distribution network is proposed that combines the linearization method and the recursive method. It can realize fast and accurate calculation of maximum DG hosting capacity in the distribution network.

- An improved comprehensive sensitivity index is proposed, which can accurately find the most suitable DG connection node for power correction during the recursive process.

- A safety constraint verification method based on safety region is proposed, which can realize rapid safety verification calculations and improve the calculation efficiency of recursive algorithm.

The remainder of this paper is organized as follows: Section 2 introduces the concepts of the distribution network operating point and the maximum DG hosting operating point. Section 3 presents the model for assessing maximum DG hosting capacity. Section 4 presents the proposed methodology, describing a two-step method that combines the linearization method and the recursive method. Section 5 presents the test system, numerical results, and discussion.

## Maximum DG hosting operating point

### Distribution network operating point

The operating point of the distribution network is represented by the matrix $W_t$ which contains the states of the load demand and DG power at time $t$.

$$W_t = \begin{bmatrix} W_{L,t} \\ W_{DG,t} \end{bmatrix} = \begin{bmatrix} (S_{L,1,t}, \cdots, S_{L,i,t}, \cdots, S_{L,n,t}) \\ (S_{DG,1,t}, \cdots, S_{DG,i,t}, \cdots, S_{DG,n,t}) \end{bmatrix} \tag{1}$$

where $W_{L,t}$ is the load consumption vector at time $t$, $W_{DG,t}$ is the DG output power vector at time $t$, $S_{L,i,t}$ is the load consumption on node $i$ at time $t$, and $S_{DG,i,t}$ is the DG output power on node $i$ at time $t$.

The power fluctuation of loads and DGs is reflected in the variation of the distribution network operating point. The distribution network operating point does not include the power output of flexible resources such as energy storage and reactive power compensation equipment. This is because loads and DGs are part of user demand, while the function of flexible resources such as energy storage and reactive power compensation equipment is to collaborate with the distribution network to provide load supply and DG accommodation services.

### Distribution network safety region

The distribution network safety region [13, 14] is defined as the set of all operating points in the state space that satisfy safety constraints.

$$\boldsymbol{\Omega}_S = \{ W_t \mid a_{1,min} \leq g_1(W_t) \leq a_{1,max}, \cdots,$$
$$a_{i,min} \leq g_i(W_t) \leq a_{i,max}, \cdots, \tag{2}$$
$$a_{m,min} \leq g_m(W_t) \leq a_{m,max} \}$$

where $\Omega_S$ is the distribution network safety region, $g_i(W_t)$ represents the value of the distribution network state variable $a_i$ when the operating point is $W_t$, $a_{i,max}$ and $a_{i,min}$ are the upper and lower limits of the distribution network state variable $a_i$, respectively.

### Critical operating point

Under normal operating conditions, the operating point $W_t$ is distributed within $\Omega_S$. Considering the maximum hosting scenario for DG, the operating point $W_t$ is located on the boundary of $\Omega_S$. This operating point is referred to as the critical operating point, where any increase in the output power of DG will lead to a violation of the safety constraints [15–19]. Assuming a constant load level, the definition of the maximum DG hosting operating point is as follows:

- When there is only one DG connection node in the distribution network, as the output power of the DG increases, the operating point gradually approaches the boundary of $\Omega_S$

and ultimately becomes the critical operating point. This critical operating point is the maximum DG hosting operating point.

• When the distribution network has multiple DG connection nodes, increasing the DG output power at different nodes can generate several different critical operating points. Among these critical operating points, the one with the largest total DG power is referred to as the maximum DG hosting operating point.

From the above analysis, it can be concluded that as the output power of DG increases, the state variable that first violates the safety constraint on the right side of Eq (2) can be regarded as a weak point under the maximum DG hosting conditions of the distribution network. Accurately identifying these weak points is beneficial for analyzing and studying the hosting capacity of the distribution network. On the other hand, when there are multiple DG connection nodes in the distribution network, how to find the maximum DG hosting operating point is key to accurately assessing the hosting capacity of the distribution network.

## Model for assessing maximum DG hosting capacity

### Distribution network model

According to Fig 1, the complex power flow equations at each node $i$ of a radial distribution network can be described as follows: $\forall i \in N$,

$$P_{i+1} = P_i - r_i(P_i^2 + Q_i^2)/V_i^2 - p_{i+1} \tag{3}$$

$$Q_{i+1} = Q_i - x_i(P_i^2 + Q_i^2)/V_i^2 - q_{i+1} \tag{4}$$

$$V_{i+1}^2 = V_i^2 - 2(r_iP_i + x_iQ_i) + (r_i^2 + x_i^2)(P_i^2 + Q_i^2)/V_i^2 \tag{5}$$

where $i$ is the node number in the distribution network, $N$ is the set of all nodes, $P_i + jQ_i$ is the line complex power flowing away from node $i$ toward node $i + 1$, $r_i + jx_i$ is the complex impedance of the line between node $i$ and node $i + 1$, $V_i$ is the voltage at node $i$, and $p_i + jq_i$ is the apparent power consumed at node $i$.

In this paper, the active power output of DG is taken as the subject of assessment for the maximum DG hosting capacity in the distribution network.

$$p_i = p_i^L - p_i^{DG}, \forall i \in N_{DG} \tag{6}$$

where $N_{DG}$ is the set of DG connection nodes, $p_i$ is the active power consumed at node $i$. $p_i^L$ is the active power consumed by the load at node $i$, and $p_i^{DG}$ is the active power output of DG at node $i$.

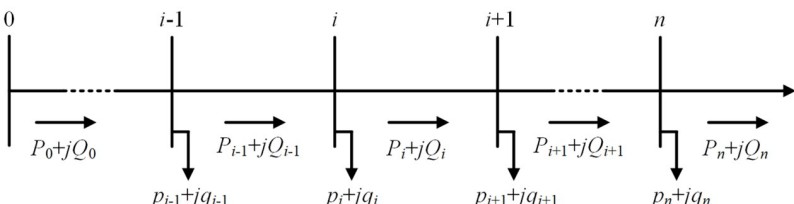

**Fig 1. Diagram of a radial distribution network with DG.**

During normal operation of the distribution network, the node voltages must satisfy voltage deviation constraints, and the line power must not exceed the overload capacity.

$$V_N - \varepsilon \leq V_i \leq V_N + \varepsilon \tag{7}$$

where $V_N$ is the the nominal voltage, $\varepsilon$ is the voltage bound.

$$P_k^2 + Q_k^2 \leq S_{k,max}^2 \tag{8}$$

where $S_{k,max}$ is the maximum apparent power allowed to flow on line $k$.

## Considerations about flexible resources

With the construction and development of new power system, the types and capacities of distributed flexible resources in distribution network are increasing. The maximum DG hosting capacity in the distribution network can be further enhanced and optimized through the scheduling of these distributed flexible resources.

**Energy storage.**   As regulating resources, energy storage (ES) can absorb and release active power through scheduling and control of charging and discharging, thereby enhancing the DG hosting capacity of the distribution network. The output power of ES should meet the following constraints.

$$p_i = p_i^L - p_i^{ES}, \forall i \in N_{ES} \tag{9}$$

$$p_{i,min}^{ES} \leq p_i^{ES} \leq p_{i,max}^{ES}, \forall i \in N_{ES} \tag{10}$$

where $N_{ES}$ is the set of nodes at which ES have been installed, $p_i^{ES}$ is the active power output of ES at node $i$, $p_{i,max}^{ES}$ and $p_{i,min}^{ES}$ denotes the upper and lower limits of ES active power at node $i$, respectively.

**Reactive power compensation equipment.**   Compared to traditional reactive power compensation devices, Static Var Compensator (SVC) has advantages such as fast response speed, high compensation accuracy, and a wide range of reactive power regulation. SVC is selected as the reactive power compensation device in this paper, and the output power of SVC must meet the following constraints.

$$q_i = q_i^L - q_i^{SVC}, \forall i \in N_{SVC} \tag{11}$$

$$q_{i,min}^{SVC} \leq q_i^{SVC} \leq q_{i,max}^{SVC}, \forall i \in N_{SVC} \tag{12}$$

where $N_{SVC}$ is the set of nodes at which SVC have been installed, $q_i^{SVC}$ is the reactive power output of SVC at node $i$, $q_{i,max}^{SVC}$ and $q_{i,min}^{SVC}$ denotes the upper and lower limits of SVC reactive power at node $i$, respectively.

## Objective function

The DG power injected into the distribution network is considered as the objective function to be maximized [20, 21].

$$f_t = \max \sum_{i \in N_{DG}} p_{i,t}^{DG} \tag{13}$$

where $f_t$ denotes the real-time assessment result of the maximum DG hosting capacity in distribution network at time t, $p_{i,t}^{DG}$ is the active power output of DG on node $i$ at time $t$.

## The two-step method

To accurately assess the maximum DG hosting capacity in the distribution network, this paper proposes a two-step method that combines the linearization method and the recursive method. Firstly, the maximum DG hosting capacity in the distribution network is preliminarily assessed using the linearization method, and then the preliminary assessment results are precisely corrected using the recursive method.

## The linearization method

To mitigate the non-linearity of the model [22], we apply linear simplification to Eqs (3), (4) and (5). Physically, the quadratic terms in these equations represent line losses, which, in the practical operation of the distribution network, are considerably smaller compared to the line power terms $P_i$ and $Q_i$. Hence, the linear simplification is as follows.

$$P_{i+1} = P_i - p_{i+1} \tag{14}$$

$$Q_{i+1} = Q_i - q_{i+1} \tag{15}$$

$$V_{i+1}^2 = V_i^2 - 2(r_i P_i + x_i Q_i) \tag{16}$$

When the voltage bound $\varepsilon$ is small enough, we can assume that $(V_i - V_N)^2 \approx 0$ and further linearize Eq (16).

$$V_i - V_{i+1} = (r_i P_i + x_i Q_i)/V_N \tag{17}$$

Although the above linearization method improves the solvability and computational efficiency of the model, it will lead to certain errors, so the errors need to be defined.

$$e = f - f_{linear} \tag{18}$$

where $e$ denotes the error index, $f$ represents the true value of the maximum DG hosting capacity of the distribution network, and $f_{linear}$ denotes the calculation result of the linearization method.

## The recursive method

To further improve the accuracy of the maximum DG hosting capacity assessment results, this paper proposes a recursive method based on improved comprehensive sensitivity and safety region models.

**Improved comprehensive sensitivity index.**   The recursive method achieves gradual correction of the assessment results of the maximum DG hosting capacity through iterative computation. Therefore, it is necessary to accurately calculate the position of the optimal power correction node at each iteration process. Based on this, this paper proposes an improved comprehensive sensitivity index that reflects the impact of DG output power on the state variables of the distribution network and accurately calculates the optimal power correction nodes. Additionally, the proposed index can identify potential weak points [23, 24].

According to Fig 2, the voltage loss of the line between node $i$ and node $i+1$ is as follows.

$$\Delta V_{i+1 \sim i} = V_{i+1} - V_i = -\frac{r_i P_i + x_i Q_i}{V_N} - j\frac{x_i P_i + r_i Q_i}{V_N} \tag{19}$$

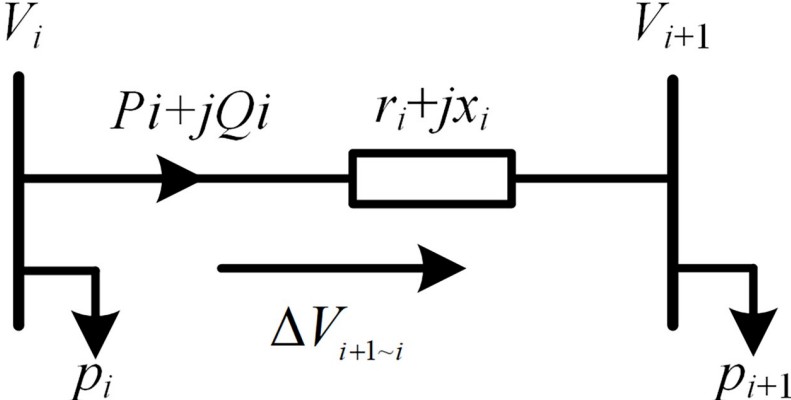

**Fig 2. Branch power flow model.**

Considering that the voltage phase angle difference between two nodes is relatively small in distribution network. Eq (19) is simplified as follows.

$$\Delta V_{i+1\sim i} \approx -\frac{r_i P_i + x_i Q_i}{V_N} \tag{20}$$

For the radial distribution network structure shown in Fig 1, the voltage at node $i$ can be expressed as follow.

$$V_i = V_N + \Delta V_{i\sim i-1} + \cdots + \Delta V_{2\sim 1} + \Delta V_{1\sim 0} \tag{21}$$

Consider the situation when DG is connected to the distribution network, Eq (21) can be rewritten as follows.

$$\Delta V_{i+1\sim i} = -\frac{r_i(P_i - aP_{DG}) + x_i Q_i}{V_N} \tag{22}$$

where $P_{DG}$ is the DG output power, $a$ is the topology coefficient, when the DG output power affects the line power between node $i$ and node $i+1$, $a=1$, otherwise, $a=0$.

The variation of $V_i$ after DG is installed at node $p$ is as follows.

$$\Delta V_{i,p} = \frac{\sum_{k\in L(i,p)}(r_k P_{DG})}{V_N} \tag{23}$$

where $r_k$ is the resistance of line $k$, $L(i, p)$ is the set of common shortest paths from node $i$ and node $p$ to the power supply.

The voltage sensitivity at node $i$ with respect to active power injection at the DG connection node $p$ is as follow.

$$\delta_{i,p} = \frac{\sum_{k\in L(i,p)} r_k}{V_N} \tag{24}$$

$\delta_{i,p}$ reflects the effect of the power change of the DG at node $p$ on the voltage at node $i$.

According to the fundamental theory of power network, for a power network composed of $n$ nodes and $m$ lines, the relationship between the currents of line and the injected current at

node is as follows.

$$I_b = AI_l \tag{25}$$

where $I_b$ is a one-column matrix that represents the injected currents at node, $I_l$ is a one-column matrix that represents the line currents, and $A$ is the node-branch incidence matrix, with its elements defined as follows.

$$a_{ij} = \begin{cases} 1 & \text{node } i \text{ is the sending node of branch } j \\ -1 & \text{node } i \text{ is the receiving node of branch } j \\ 0 & \text{otherwise} \end{cases} \tag{26}$$

Eq (25) can be rewritten as follows.

$$I_l = C(\lambda)I_b \tag{27}$$

where $C$ is the inverse matrix of $A$, and the elements of $C$ are constants. The calculation of the elements in $I_l$ is as follows.

$$I_{l,h} = \lambda_{h1}I_{b,1} + \cdots + \lambda_{hi}I_{b,i} + \cdots + \lambda_{hn}I_{b,n} \tag{28}$$

where $I_{l,h}$ is the current of line $h$, $I_{b,i}$ is the injected current at node $i$, and $\lambda_{hi}$ is the element in the $h^{th}$ row and $i^{th}$ column of matrix $C$.

Eq (28) can be rewritten as follows.

$$S_{l,h} = V_{l,h} \sum_{i=1}^{n} \frac{\lambda_{hi}(P_{b,i} + jQ_{b,i})}{V_{b,i}} \tag{29}$$

where $S_{l,h}$ is the apparent power of line $h$, $V_{l,h}$ is the voltage at the starting node of line $h$, and $V_{b,i}$ is the voltage at node $i$.

The variation of $S_{l,h}$ after DG is installed at node $p$ is as follows:

$$\Delta S_{l,h} = -V_{l,h} \frac{\lambda_{hp}P_{DG}}{V_{b,p}} \tag{30}$$

The magnitude variation of $S_{l,h}$ is as follows

$$\Delta|S_{l,h}| = |S_{l,h} + \Delta S_{l,h}| - |S_{l,h}| \tag{31}$$

The line power sensitivity of line $h$ with respect to active power injection at the DG connection node $p$ is as follow.

$$\gamma_{h,p} = \frac{\lambda_{hp}P_{l,h}}{\sqrt{(P_{l,h})^2 + (Q_{l,h})^2}} \tag{32}$$

$\gamma_{h,p}$ reflects the effect of the power variation at node $p$ on the apparent power magnitude of line $h$.

In order to find the most suitable DG connection node for power correction during the recursive process, this paper proposes an improved comprehensive sensitivity index. The rationale of the improved comprehensive sensitivity is as follows.

The improved node voltage sensitivity is obtained from Eq (24).

$$\delta'_{i,p} = \frac{\delta_{i,p}}{V_{lim} - V_{i,t}} \tag{33}$$

where $V_{lim}$ represents the voltage limit, and $V_{i,t}$ is the voltage at node $i$ at time t.

The improved line power sensitivity is obtained from Eq (32).

$$\gamma'_{h,p} = \frac{\gamma_{h,p}}{S_{lim} - S_{h,t}} \tag{34}$$

where $S_{lim}$ represents the line power limit, $S_{h,t}$ is the power of line $h$ at time $t$.

The improved comprehensive sensitivity of DG connection node $p$ is as follows:

$$\xi_p = \max\left(\delta'_{i_1,p}, \cdots, \delta'_{i_n,p}, \gamma'_{h_1,p}, \cdots, \gamma'_{h_k,p}\right), p \in N_{DG} \tag{35}$$

Under the optimization objective represented in Eq (13), when the recursive process requires an increase in DG output power, choose the DG connection node with the smallest $\xi$ as the power correction node; when the recursive process requires a decrease in DG output power, choose the DG connection node with the largest $\xi$ as the power correction node. In addition, it can be seen from Eq (35) that the value of $\xi_p$ is obtained from the sensitivity of a certain state variable. This suggests that as the DG output power increases at node $p$, this state variable will reach the safety boundary first. Therefore, the improved sensitivity index proposed in this paper can assist in identifying weak points in the distribution network.

**Safety constraint verification based on safety region.** The recursive method requires safety constraint verification of the corrected power to determine whether to terminate the recursive iteration process. Traditional methods typically use commercial software for power flow calculations to achieve safety constraint verification. To improve computational efficiency, a safety constraint verification method based on the safety region model is proposed.

Referring to Eq (2), the safety region of an active distribution network can be represented as follows.

$$-\boldsymbol{c} \leq \boldsymbol{\alpha} A \Delta \boldsymbol{W}_t + \boldsymbol{S}_t \leq \boldsymbol{c} \tag{36}$$

Eq (36) represents the line power safety region, which can realize the safety constraint verification of the line power. $\boldsymbol{c}$ is the column vector of the power boundary, and the power safety boundary includes positive and negative boundaries, which indicates that bidirectional power flow may occur in the lines of active distribution networks. $\boldsymbol{\alpha}$ is the constraint coefficient matrix, which can be calculated through the network loss coefficient. The larger the values of the elements in $\boldsymbol{\alpha}$, the stricter the safety constraints of the line power. The elements $a_{ij}$ in matrix $A$ represent whether the $i$th state variable is within the $j$th constraint: if so, the value is 1; otherwise, it is 0. $\Delta \boldsymbol{W}_t$ is the correction amount of $\boldsymbol{W}_t$. $\boldsymbol{S}_t$ represents the line power when the distribution network operating point is $\boldsymbol{W}_t$.

$$\boldsymbol{V}^- \leq \boldsymbol{\beta} B \Delta \boldsymbol{W}_t + \boldsymbol{V}_t \leq \boldsymbol{V}^+ \tag{37}$$

Eq (37) represents the node voltage safety region, which can realize the safety constraint verification of the node voltage. $\boldsymbol{V}^+$ and $\boldsymbol{V}^-$ are the column vectors of the upper and lower boundaries of the voltage, respectively. $\boldsymbol{\beta}$ is the constraint coefficient matrix, and the larger the values of the elements in $\boldsymbol{\beta}$, the stricter the constraints of the node voltage. The coefficient matrix $B$ can be calculated by Eq (24). $\boldsymbol{V}_t$ represents the node voltage when the distribution network operating point is $\boldsymbol{W}_t$.

**The framework of the two-step method.** The steps of the two-step method are as follows.

Step 1: Use the linearization method to calculate the preliminary assessment result of the maximum DG hosting capacity in distribution network. Calculate the preliminary DG maximum hosting operating point $W_t$ and the corresponding state variables $V_t$, $S_t$.

Step 2: Use power flow calculation to determine whether the distribution network operating point $W_t$ is within the safety constraints. If so, proceed to step 3; if not, proceed to step 5.

Step 3: Calculate the improved comprehensive sensitivity index $\xi$ for all DG connection nodes in the distribution network based on Eq (35). Increase the DG output power slightly at the node with the smallest $\xi$ and update the distribution network operating point $W_t$.

Step 4: Determine whether the updated distribution network operating point $W_t$ is within the safety region as indicated in Eqs (36) and (37). If so, return to Step 3; if not, proceed to Step 7.

Step 5: Calculate the improved comprehensive sensitivity $\xi$ for all DG connection nodes in the distribution network based on Eq (35). Decrease the DG output power slightly at the node with the biggest $\xi$ and update the distribution network operating point $W_t$;

Step 6: Determine whether the updated distribution network operating point $W_t$ is within the safety region as indicated in Eqs (36) and (37). If so, proceed to Step 7; if not, return to Step 5.

Step 7: Calculate and output the corrected maximum DG hosting capacity in the distribution network.

The flow chart of the two-step method is shown in Fig 3.

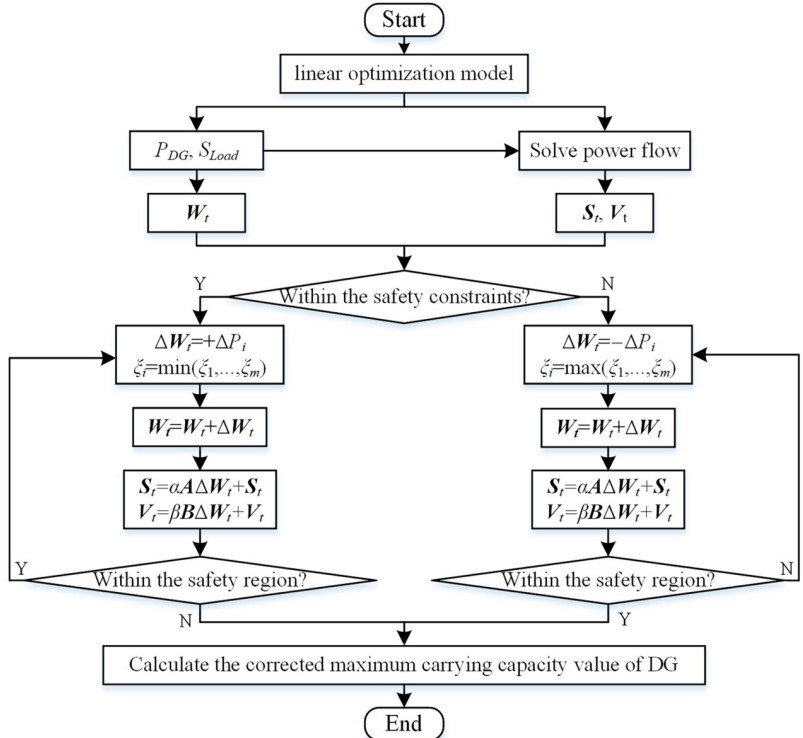

**Fig 3. Flow chart of the two-step method.**

## Case study

### Test system

In order to verify the validity of the methods proposed in this paper, the simulation example adopts the IEEE-33 bus distribution network as shown in Fig 4. The nominal power of the distribution network is 1 MVA, and the nominal voltage is 10.5 kV.

Fig 5 shows the daily load consumption profiles.

The assessment of the maximum DG hosting capacity in distribution network is mainly used in safety boundary analysis and DG capacity planning. Two scenarios were set up for

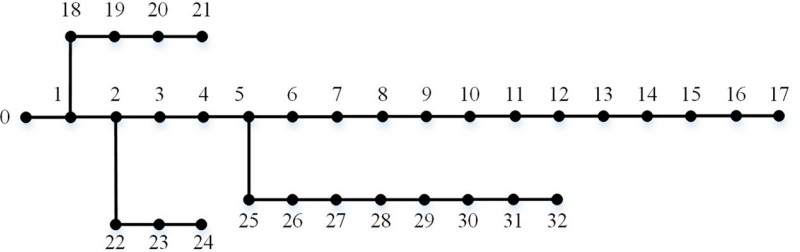

**Fig 4. Diagram of IEEE-33 bus distribution network.**

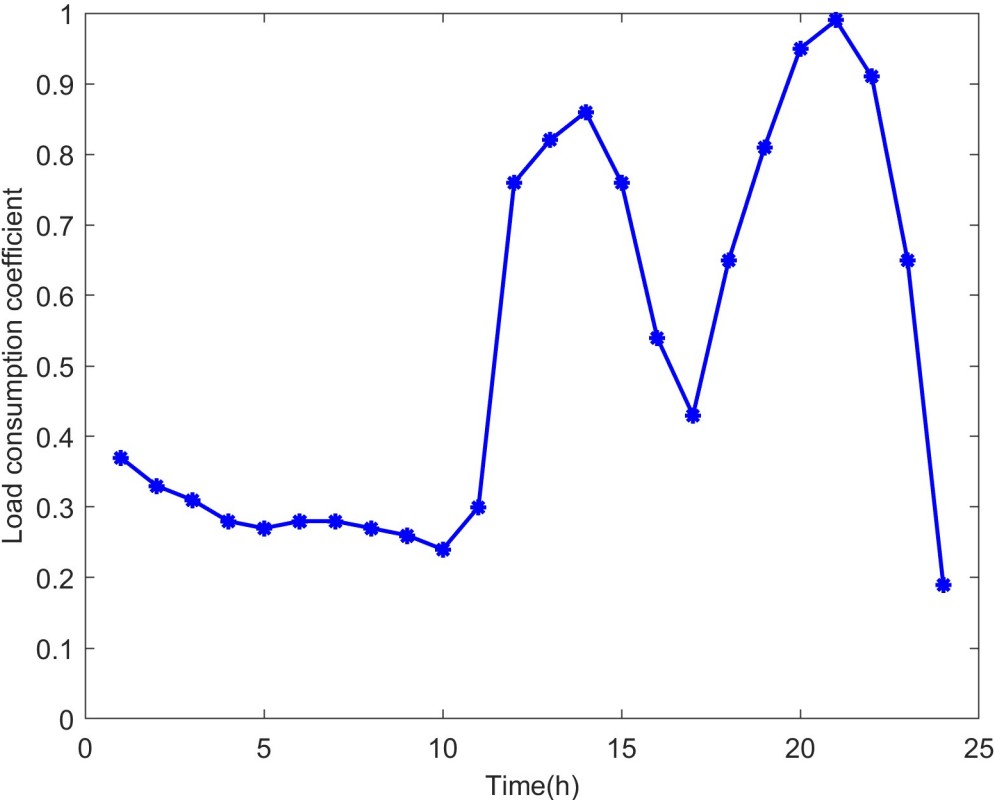

**Fig 5. Load consumption profiles.**

simulation verification. Scenario 1 mainly considers the nodes where DG has been installed, while Scenario 2 additionally considers the nodes where DG is planned to be installed.

Scenario 1: The nodes where DG has been installed are nodes 15 and 29. ES is installed at nodes 17 and 32, each with an installation capacity of 300kVA. SVC is installed at node 24, offering a continuously adjustable reactive power output ranging from -500 kVar to 500 kVar.

Scenario 2: The nodes where DG has been installed are nodes 15 and 29. The nodes 8, 20, and 22 are planned for DG installation. Similar to Scenario 1, ES is installed at nodes 17 and 32, with installation capacities of 300kVA. SVC is installed at node 24, with a reactive power output vary from -500 kVar to 500 kVar.

The upper and lower limits of the node voltage of the distribution network are set to 1.07 p. u. and 0.93 p.u. The thermal limit of the line is set to 2 MVA.

## Simulation and analysis

**Real-time assessment of maximum DG hosting capacity.** The simulation results of the real-time assessment method of maximum DG hosting capacity in distribution network are shown in Figs 6 and 7.

According to the analysis of the safety boundary and critical operating point of the distribution network, it can be seen that the maximum DG hosting capacity is related to the load level

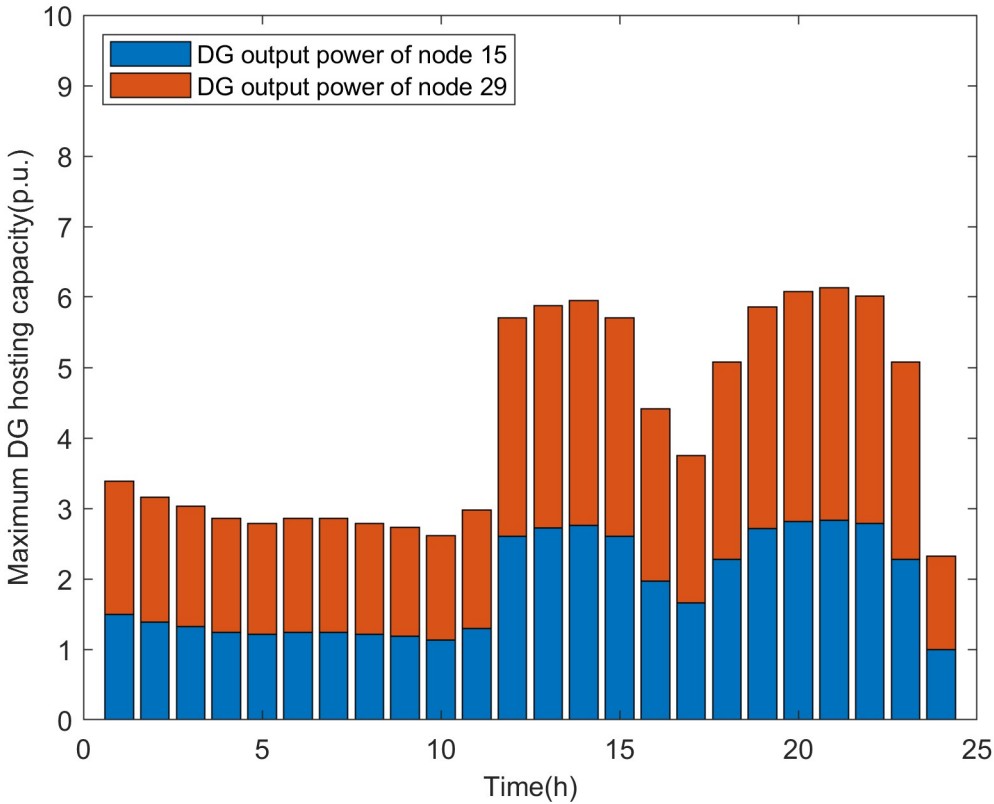

**Fig 6. The real-time assessment results of the maximum DG hosting capacity in scenario 1.**

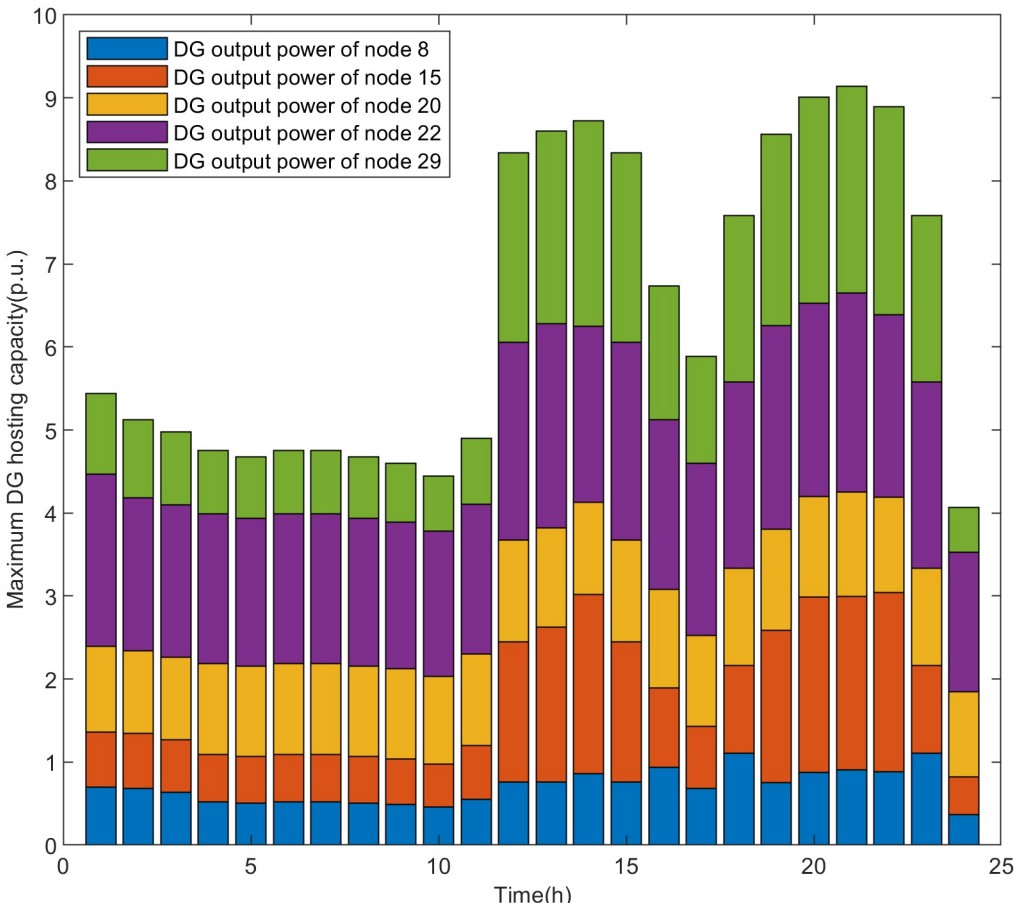

**Fig 7. The real-time assessment results of the maximum DG hosting capacity in scenario 2.**

of the distribution network. The simulation results shown in Figs 6 and 7 are basically consistent with the changing trend of the load curve shown in Fig 5. The real-time assessment results of the maximum DG hosting capacity in the distribution network can help us more accurately understand the safety boundaries and operational limits of the distribution network. It also provides data references and theoretical support for the future expansion and upgrading of DG.

**Operating boundary and weak points.** Based on the assessment results, the maximum DG hosting operating point is obtained. The distribution network state variables at the maximum DG hosting operating point for each scenario are as follows.

It can be seen from Figs 8 and 9 that with fewer DG connection nodes, the increase in DG output power mainly leads to the rise in node voltage. The voltages of nodes 15 and 29 are the state variables closest to the safety boundary, indicating that the voltages at nodes 15 and 29 are the weak points under the maximum DG hosting condition. This means that if the DG output power increases further, the voltages at these nodes are most likely to violate the safety constraints.

It can be seen from Figs 10 and 11 that the increase in DG output power in the distribution network not only leads to a rise in node voltage but also alters the power flow direction of the lines. This change may potentially cause reverse power overload in the lines if the DG output

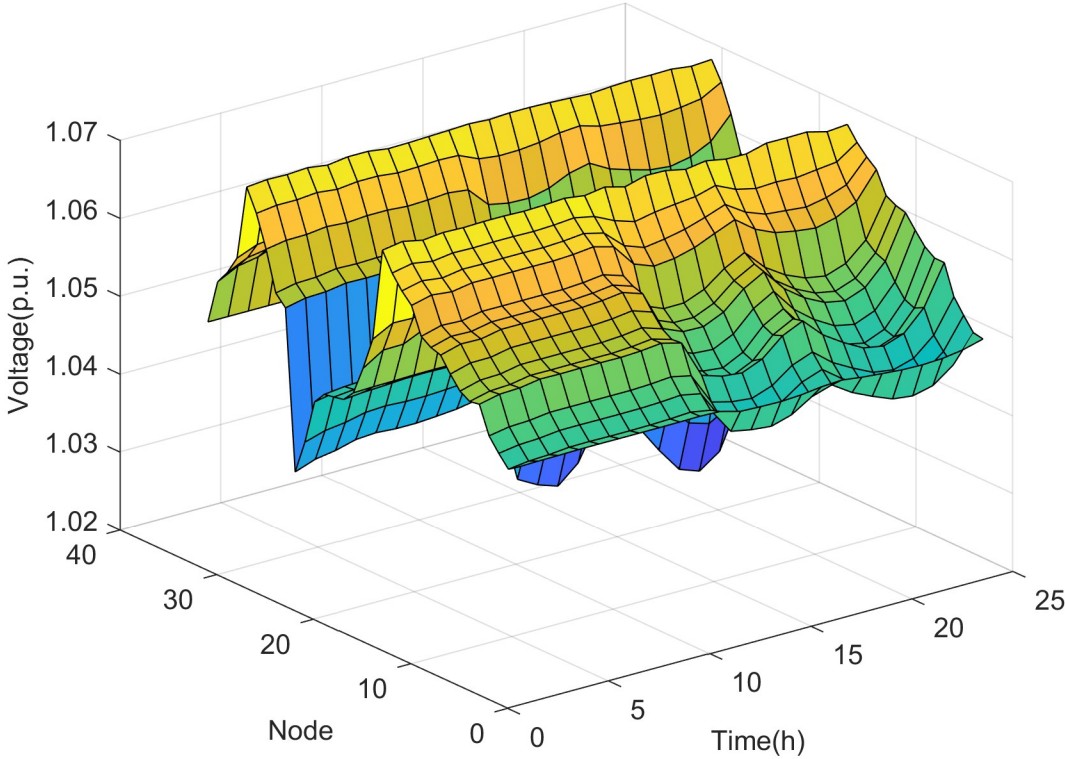

**Fig 8. The voltage of nodes in scenario 1.**

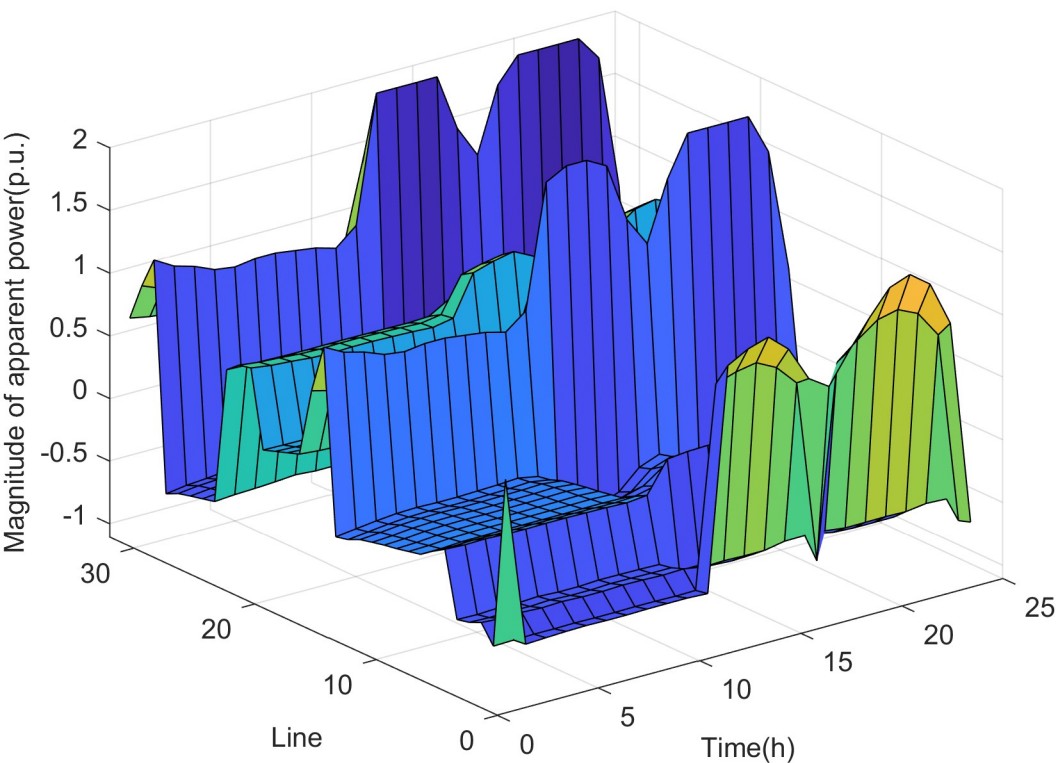

**Fig 9. The power magnitude of lines in scenario 1.**

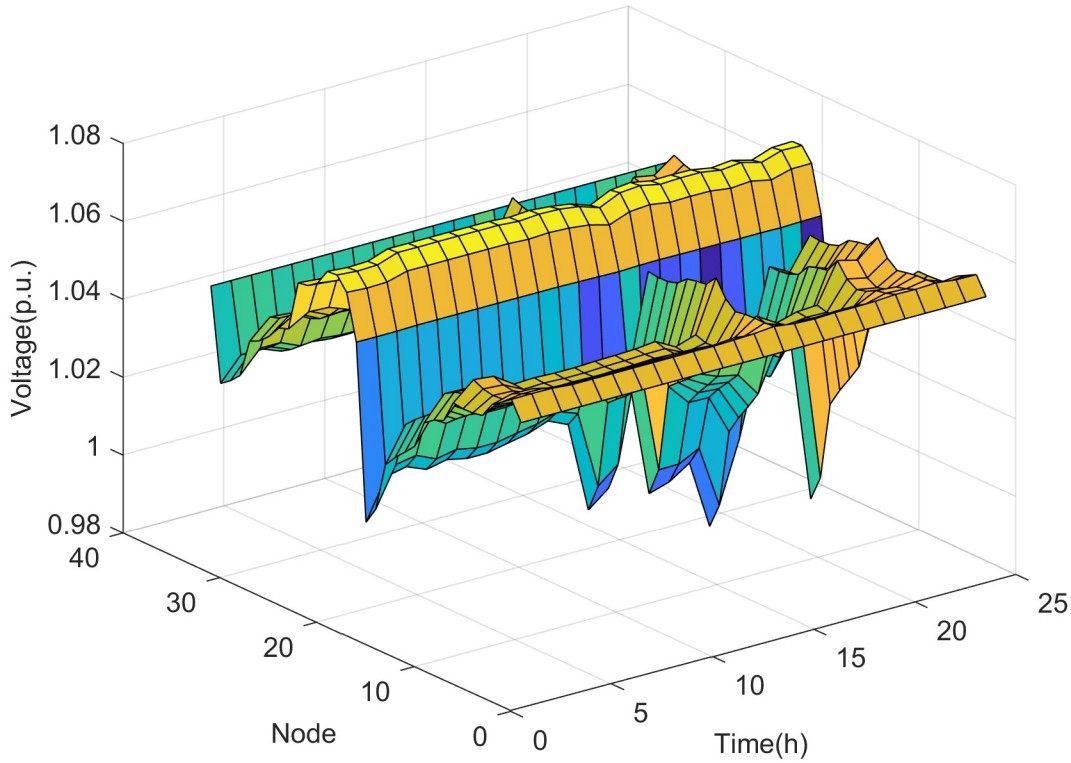

**Fig 10. The voltage of nodes in scenario 2.**

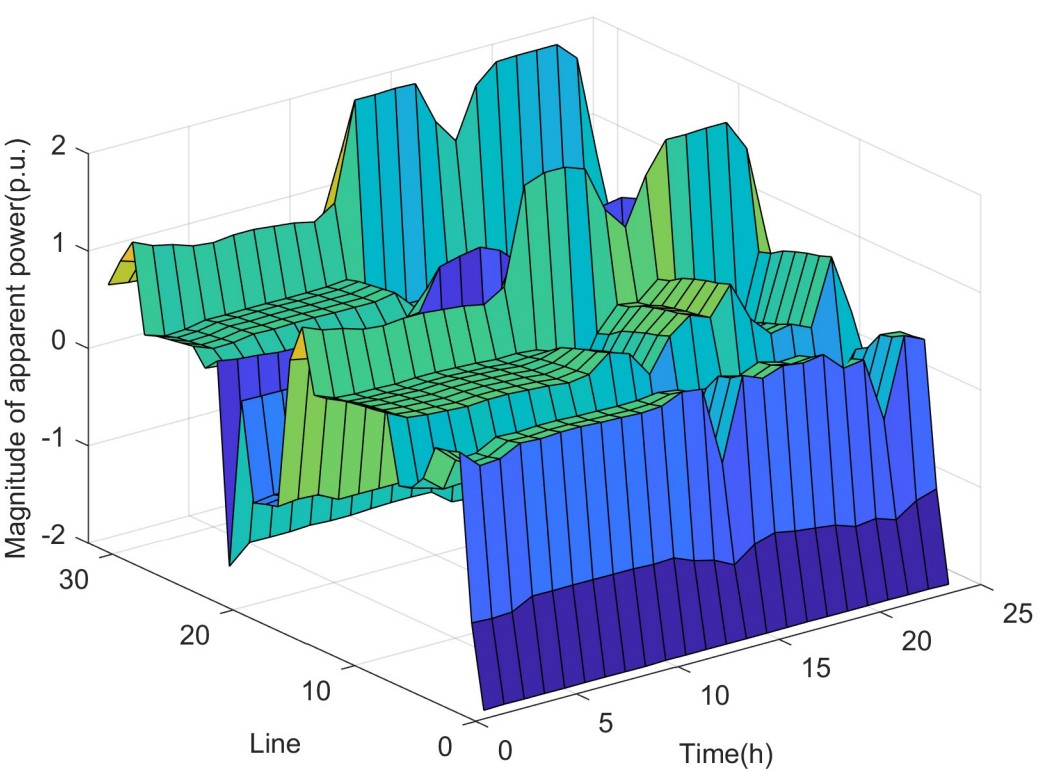

**Fig 11. The power magnitude of lines in scenario 2.**

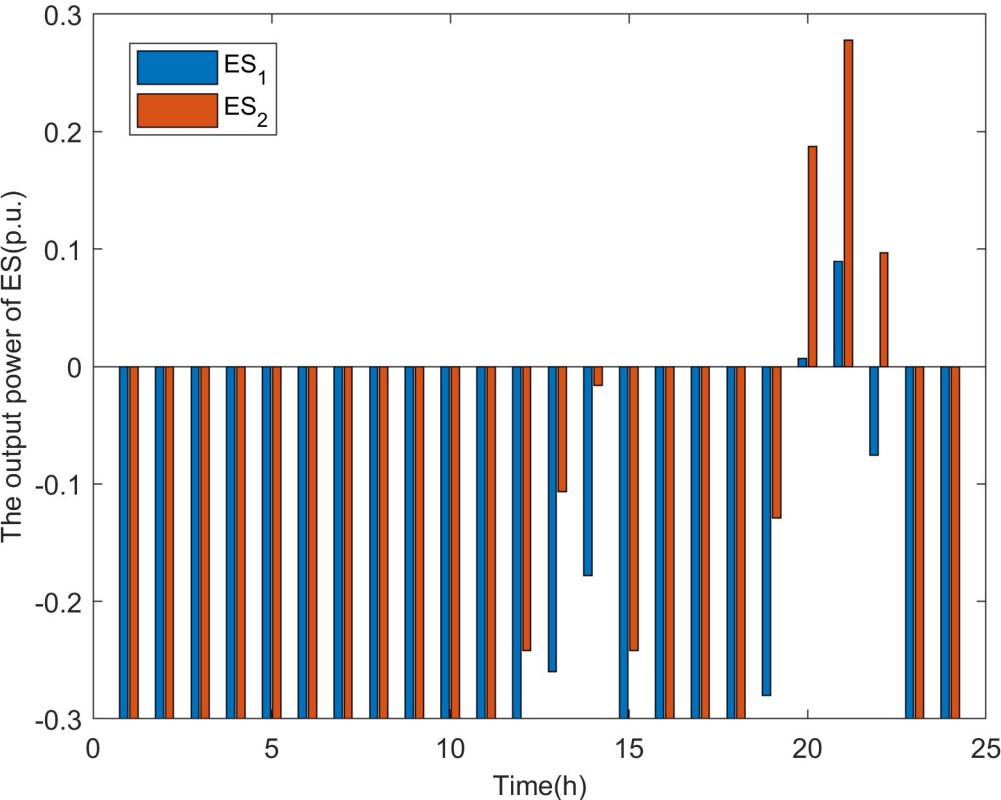

**Fig 12. The output power of ES in scenario 1.**

power reaches a sufficiently high level. As shown in Fig 11, the power of the main feeder (the line between node 0 and node 1) is the state variable closest to the safety boundary in scenario 2. This indicates that the power of the main feeder is the weak point under the maximum DG hosting condition, meaning that if the DG output power increases further, the power on the main feeder is most likely to violate the safety constraints.

**The output power of ES and SVC.** The results of the output power of flexible resources such as ES and SVC are as follows.

It can be seen from Figs 12 and 13 that, the operating mode of the ES varies with the load level of the distribution network. In Scenario 1, when the load level is high (from 20:00 to 22:00), the ES discharges to reduce the load consumption at nodes 17 and 32, ensuring that the power on $line_{15}$ and $line_{29}$ does not exceed safe levels. When the load level is low (from 00:00 to 19:00, from 23:00 to 24:00), as previous research has shown, the voltage at the DG connection nodes becomes the weak point when there are fewer DG connection nodes. Therefore, to allow the distribution network to host more DG power, the ES charges to reduce the voltage rise at DG connection nodes caused by increased DG power. Similarly, the SVG operates as a reactive load, absorbing reactive power to lower the voltage levels at the nodes.

As shown in Fig 14, in Scenario 2, when the load level is high (from 20:00 to 22:00), the ES discharges to reduce the line power of $line_{15}$ and $line_{29}$. When the load level is low (from 00:00 to 19:00, from 23:00 to 24:00), as previous research has indicated, the line power of the main feeder becomes the weak point when there are multiple DG connection nodes. Therefore, to

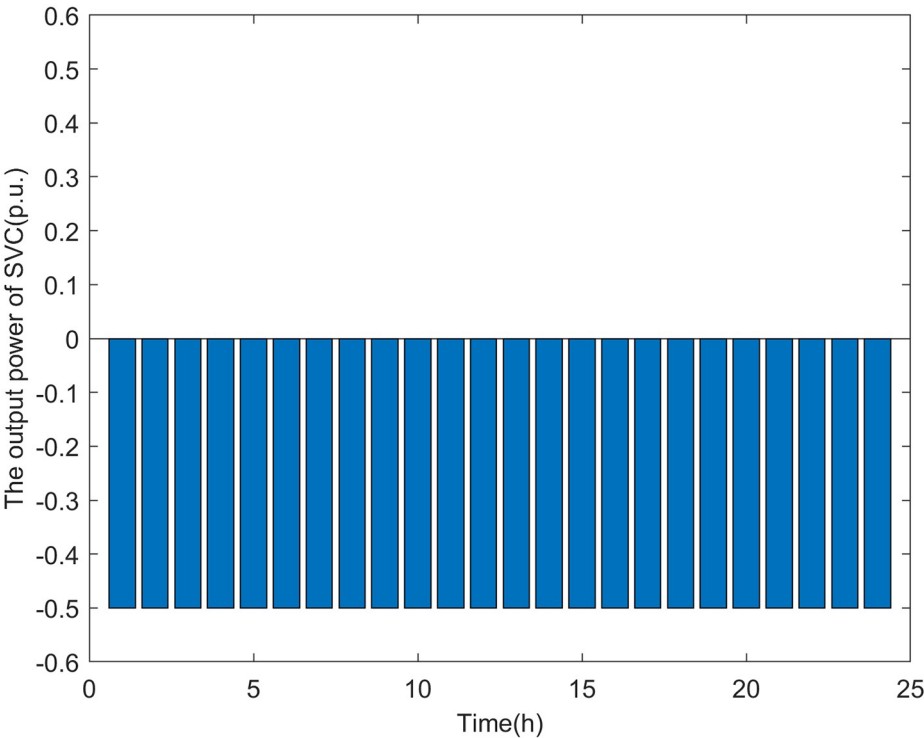

**Fig 13. The output power of SVC in scenario 1.**

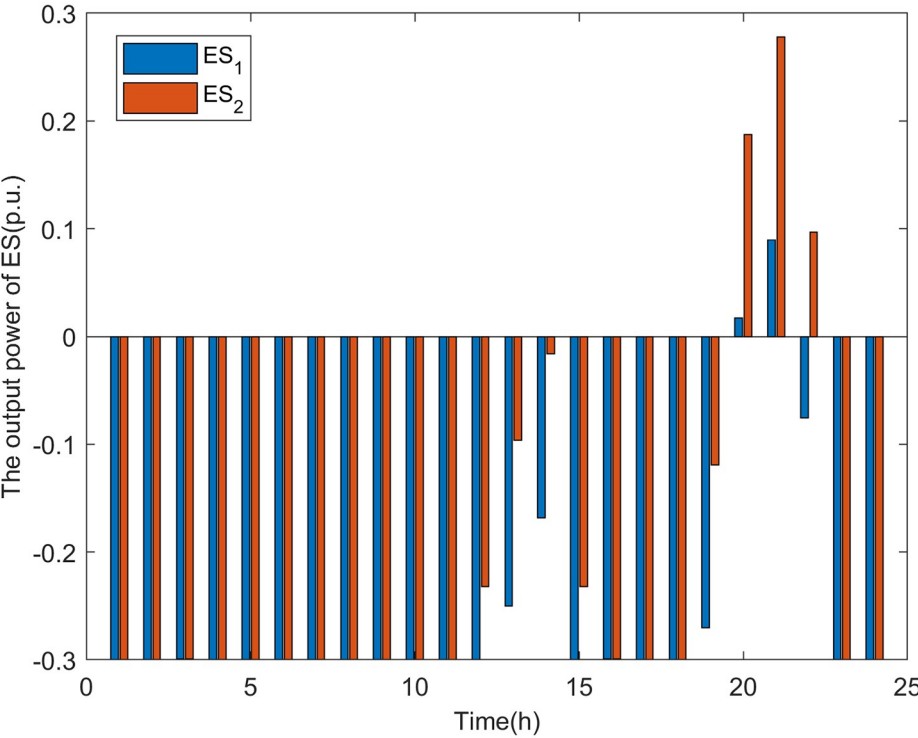

**Fig 14. The output power of ES in scenario 2.**

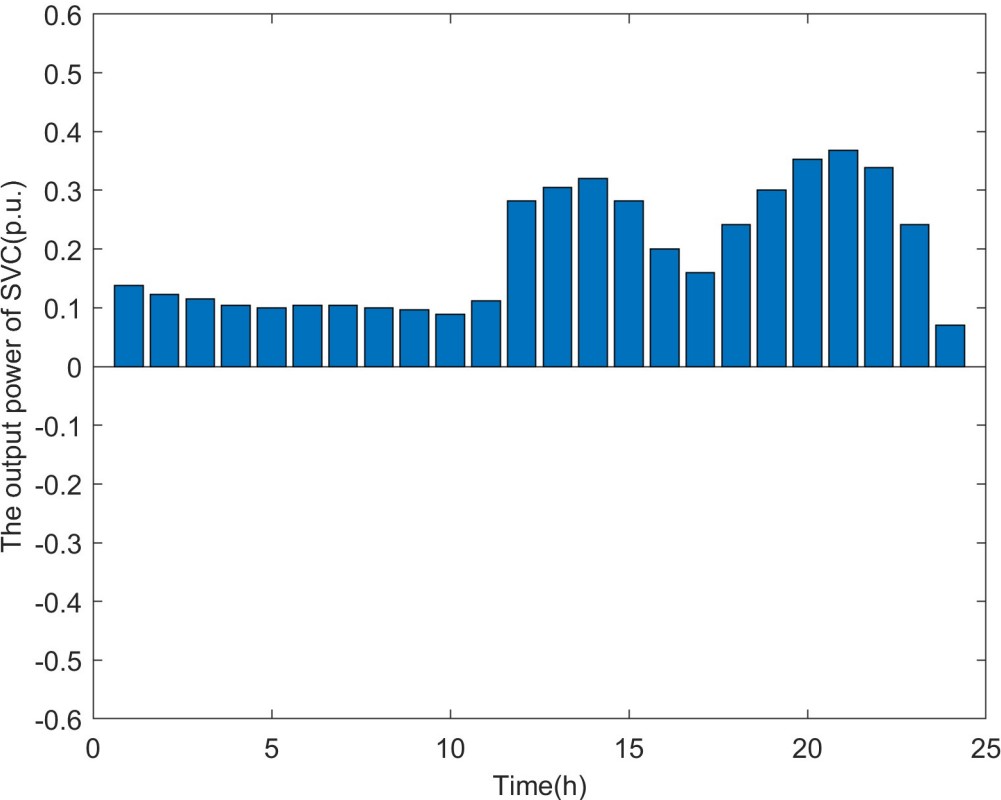

**Fig 15. The output power of SVC in scenario 2.**

allow the distribution network to host more DG power, the ES operates in charging mode to mitigate the increase in reverse power on the main feeder caused by the rise in DG output power.

As shown in Fig 15, the SVC operates as a reactive power source in Scenario 2. As previous studies have shown, the line power of the main feeder becomes the weak point when there are multiple DG connection nodes. The line power of the main feeder mainly consists of reverse active power and forward reactive power. The former is caused by reverse power transmission due to the increase in DG output, while the latter is due to the reactive power load consumption of the distribution network. To enable the distribution network to host more DG power, the SVC functions as a reactive power source to compensate for the reactive power load in the network, thereby reducing the reactive component of the apparent power on the main feeder.

**Validity verification of simulation results.**   The correction of the preliminary assessment results of the maximum DG hosting capacity in the two scenarios using the proposed recursive method is shown in Figs 16 and 17.

Comparison is made between the proposed two-step method and the linear optimization method commonly used in other literature. The state variables of the distribution network are calculated based on the maximum DG hosting operating points obtained by both methods. The distance between the operating state of the distribution network and the safety boundary

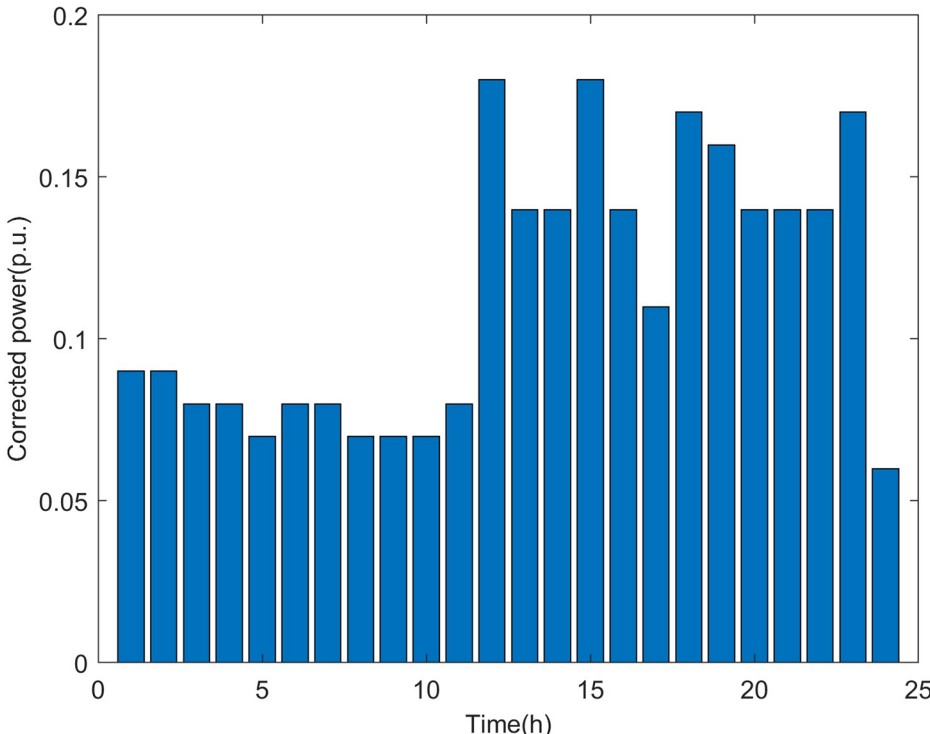

**Fig 16. Corrected power of the maximum DG hosting capacity in scenario 1.**

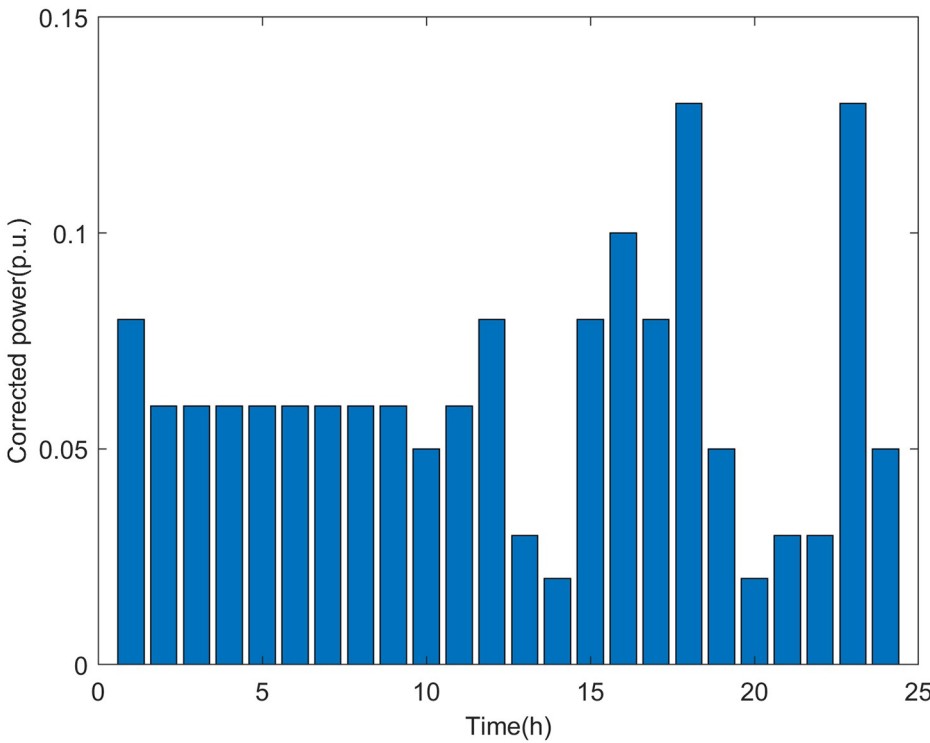

**Fig 17. Corrected power of the maximum DG hosting capacity in scenario 2.**

is evaluated using $D_V$ and $D_S$, which are defined as follows:

$$D_{S,t} = \frac{\sum\limits_{h=1}^{k} |S_{lim} - S_{h,t}|}{k} \tag{38}$$

where $k$ is the number of lines in the distribution network, $S_{lim}$ represents the line power limit, and $S_{h,t}$ is the power of line $h$ at time $t$.

$$D_{V,t} = \frac{\sum\limits_{i=1}^{n} |V_{lim} - V_{i,t}|}{n} \tag{39}$$

where $n$ is the number of nodes in the distribution network, $V_{lim}$ represents the voltage limit, and $V_{i,t}$ is the voltage at node $i$ at time $t$.

It can be seen from Eqs (38) and (39) that the smaller the values of $D_V$ and $D_S$, the closer the operating state of the distribution network is to the safety boundary, indicating higher accuracy in the assessment results.

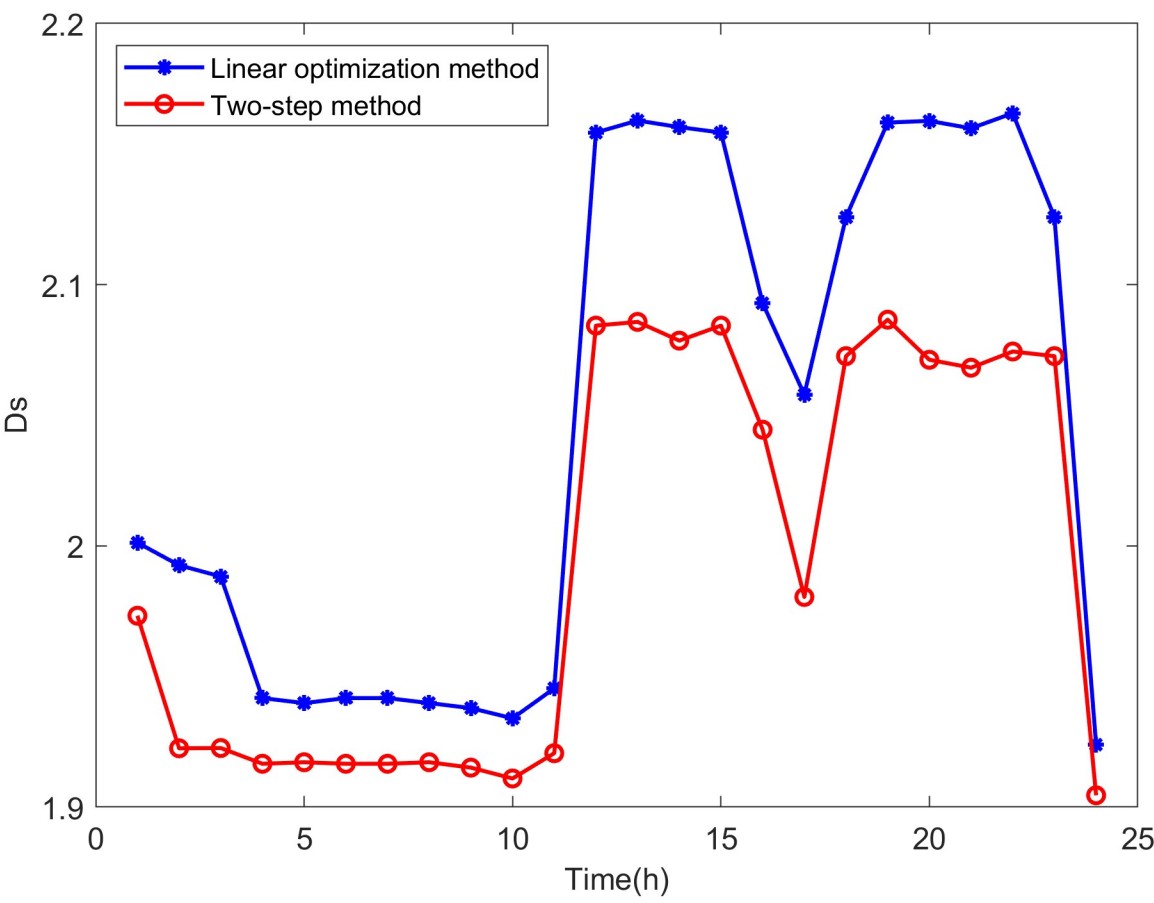

**Fig 18. The $D_S$ in scenario 1.**

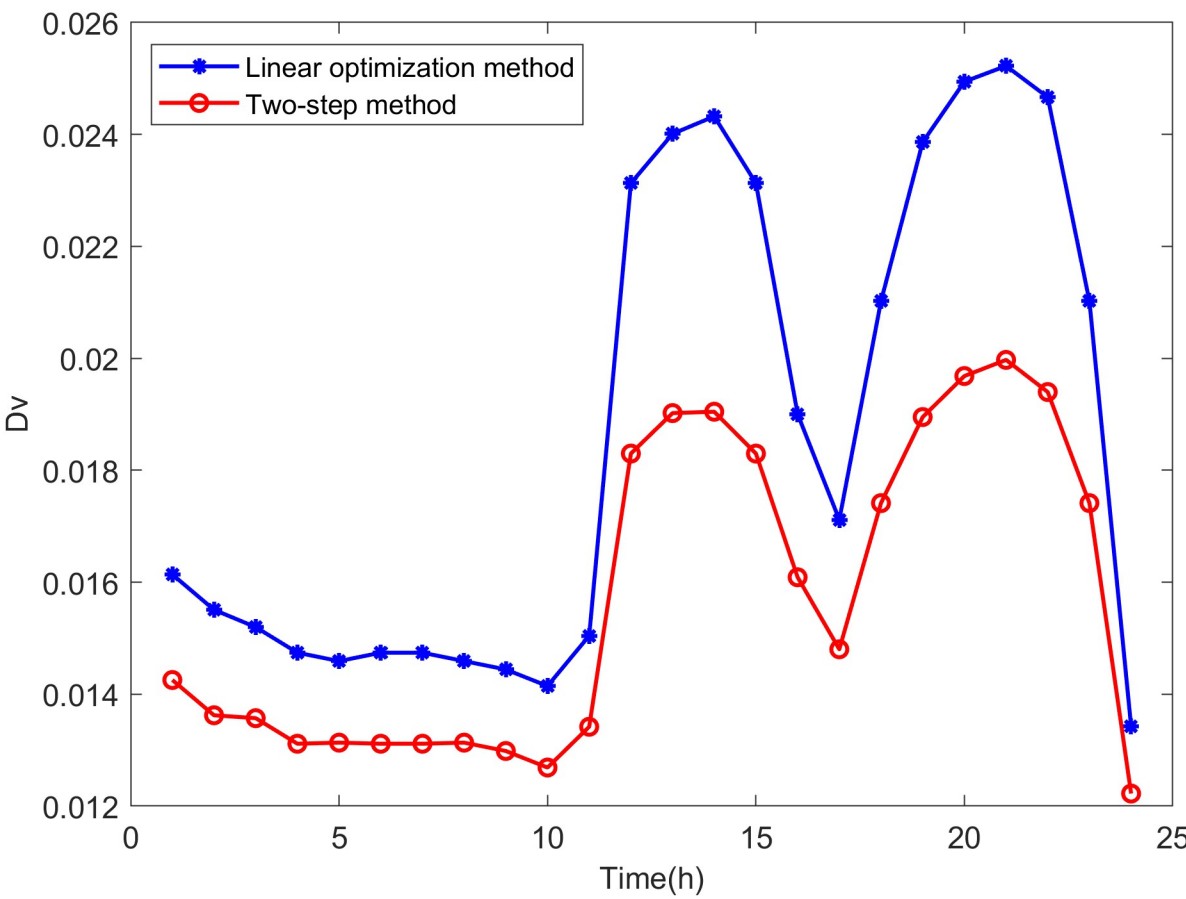

**Fig 19. The $D_V$ in scenario 1.**

From Figs 18–21, it can be seen that the maximum DG hosting operating point obtained by the two-step method has lower $D_V$ and $D_S$ values, indicating that the proposed two-step method provides more accurate assessment results, whereas the results from the linear optimization method are more conservative.

**Comparison of different sensitivity indexes.**   This subsection compares the performance of different sensitivity indexes in searching for the most suitable DG connection node for power correction. The steps of searching for power correction node using traditional sensitivity index are as follows:

Step 1: Find the state variable $var_i$ with the smallest difference from the limit.

Step 2: Calculate the sensitivity of $var_i$ with respect to the injected power at each DG connection node according to Eqs (24) and (32).

Step 3: Based on the traditional sensitivity index in step 2, determine the DG connection node for power correction.

The following index $\sigma$ is used to evaluate the performance improvement of different sensitivity indexes in enhancing the accuracy of the maximum DG hosting capacity assessment

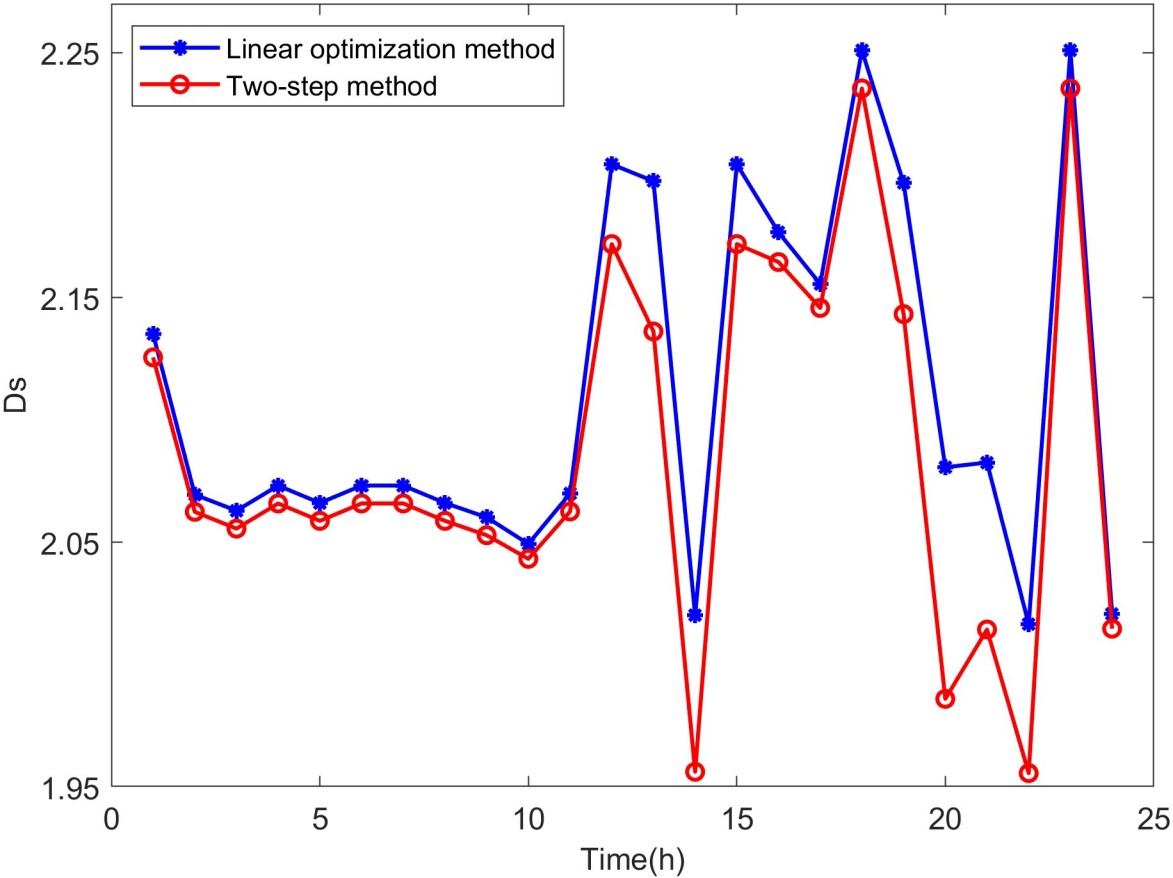

**Fig 20. The $D_S$ in scenario 2.**

results.

$$\sigma = \frac{\sum_{t \in T} \Delta P_{DG,t}}{T} \tag{40}$$

where $T$ is the evaluation period, $\Delta P_{DG,t}$ is the corrected power for the maximum DG hosting capacity at time $t$. It can be seen from Eq (40) that the larger $\sigma$ is, the greater the corrected power is, indicating a higher accuracy of the calculation results.

The DG connection locations in the distribution network are obtained through random selection. The average values of $\sigma$ for the improved comprehensive sensitivity index and traditional sensitivity index under different numbers of DG connection nodes are shown as follows.

It can be seen from Fig 22 that the improved comprehensive sensitivity index proposed in this paper significantly enhances the accuracy of maximum DG hosting capacity assessment compared to the traditional sensitivity index.

## Conclusion

To accurately assess the maximum DG hosting capacity in the distribution network, this paper proposes an two-step method that combines the linearization method and the recursive

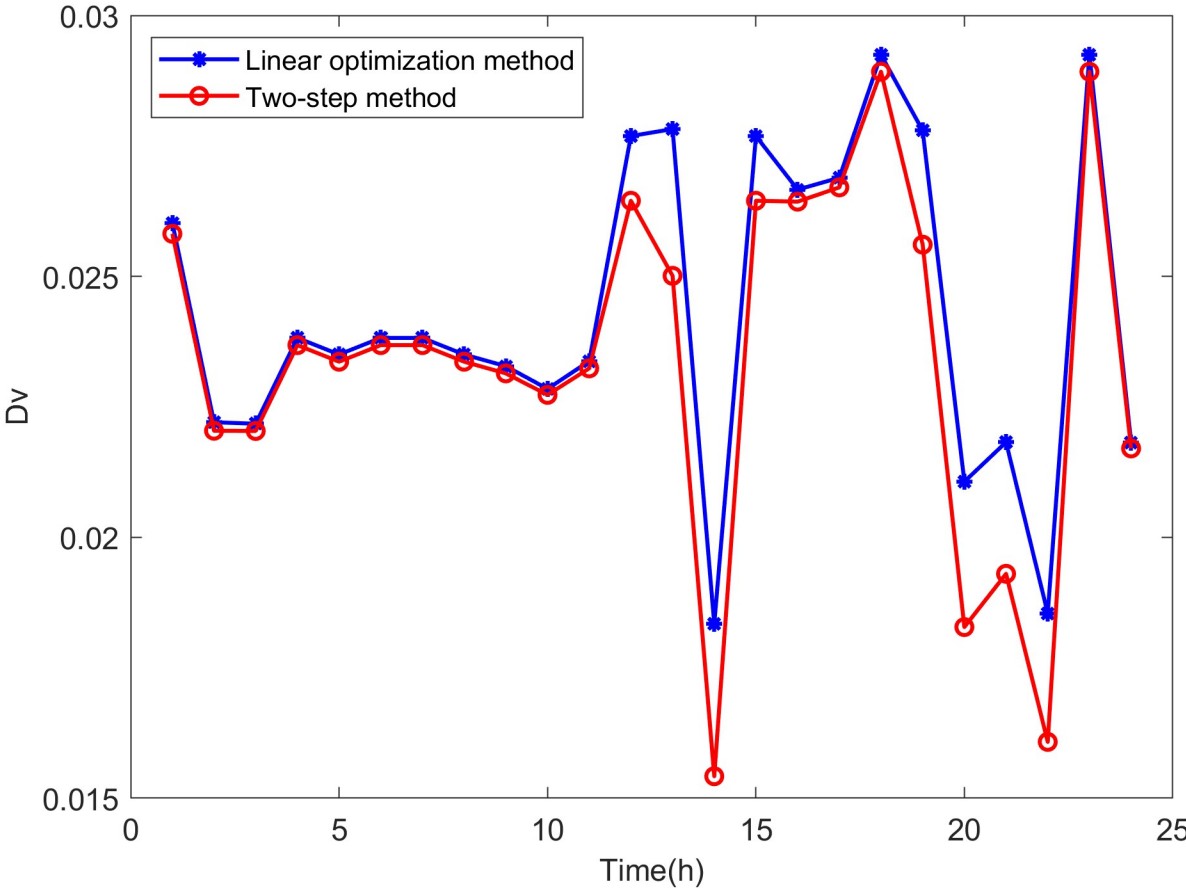

**Fig 21. The $D_V$ in scenario 2.**

method. Firstly, the maximum DG hosting capacity in the distribution network is preliminarily calculated using linear optimization method, and then the preliminary calculation results are precisely corrected using recursive method. During the recursive process, an improved comprehensive sensitivity index is proposed to find the most suitable DG connection node for power correction. The improved comprehensive sensitivity index can accurately reflect the impact of different DG connection nodes on the distribution network state variables and achieve precise identification of weak points. Additionally, this paper proposes a safety constraint verification method based on security regions, which enhances the calculation speed of the recursive algorithm. Finally, the proposed method was tested and validated on the IEEE 33-bus distribution network. The simulation results indicate that the proposed method can accurately assess the maximum DG hosting capacity in the distribution network. The improved comprehensive sensitivity exhibits significant performance improvement over traditional sensitivity index in enhancing the accuracy of calculating the maximum DG hosting capacity. To further enhance the robustness of the assessment results, the next step of this study will focus on researching an evaluation method for DG hosting capacity in distribution networks that considers various uncertainties.

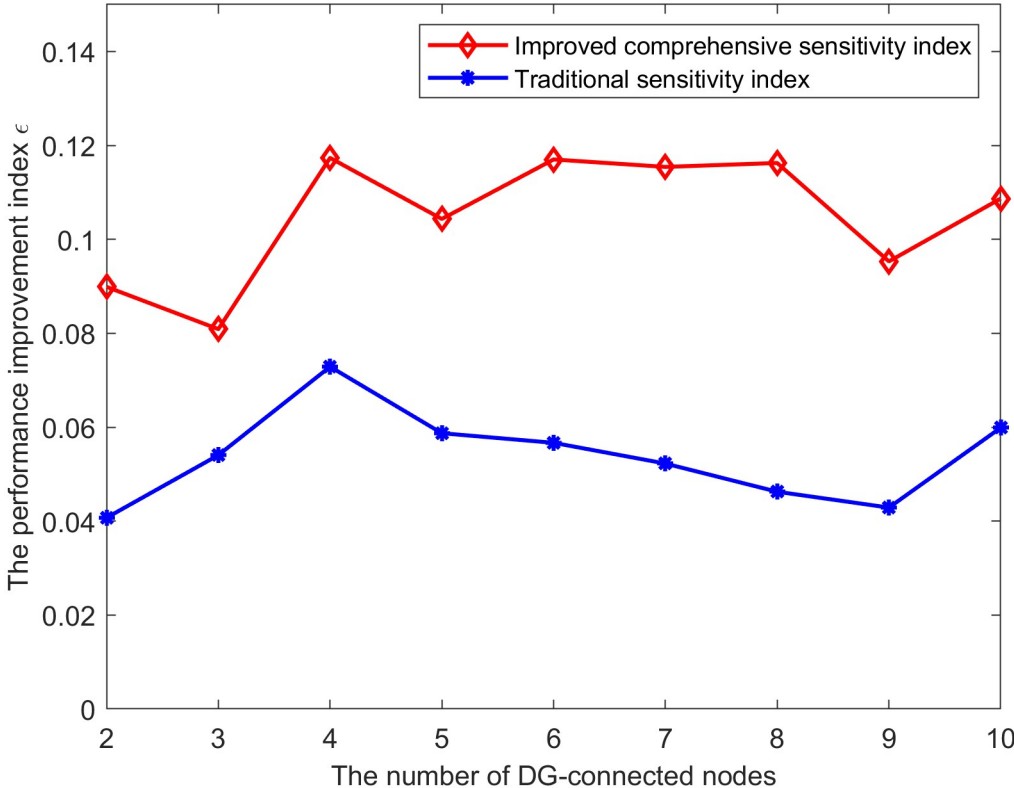

**Fig 22. The average value of $\sigma$ under different numbers of DG connection nodes.**

## Supporting information

**S1 Data.**
(XLSX)

## Author Contributions

**Conceptualization:** Dai Wan, Kailun Fan, Jingyu He.

**Data curation:** Dai Wan, Kailun Fan, Jingyu He.

**Formal analysis:** Gexing Yang.

**Investigation:** Gexing Yang.

**Methodology:** Dai Wan, Jingyu He, Haochong Zhang.

**Project administration:** Xujin Duan.

**Supervision:** Xujin Duan.

**Validation:** Kailun Fan, Haochong Zhang, Gexing Yang, Xujin Duan.

**Writing – original draft:** Dai Wan, Jingyu He, Haochong Zhang.

**Writing – review & editing:** Dai Wan.

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
