## [Decision Letter · Decision Letter 0]

8 Feb 2024

PONE-D-24-01853Research on Assessment Method of Maximum Carrying Capacity of Distributed Generation in Distribution System With High Shares of Renewables and Power ElectronicsPLOS ONE

Dear Dr. He,

Thank you for submitting your manuscript to PLOS ONE. After careful consideration, we feel that it has merit but does not fully meet PLOS ONE’s publication criteria as it currently stands. Therefore, we invite you to submit a revised version of the manuscript that addresses the points raised during the review process.

**ACADEMIC EDITOR: ** Be sure to:I just received the comments from one reviewer.Please carefully address the comments and improve the quality of the manuscipt in both English and presentations. Please clearly highlight the novelty of this manuscript.  ==============================

We look forward to receiving your revised manuscript.

Kind regards,

Jianguo Wang, PhD

Academic Editor

PLOS ONE

Journal Requirements:

Reviewers' comments:

Reviewer's Responses to Questions

**Comments to the Author**

1. Is the manuscript technically sound, and do the data support the conclusions?

Reviewer #1: Yes

2. Has the statistical analysis been performed appropriately and rigorously? 

Reviewer #1: Yes

3. Have the authors made all data underlying the findings in their manuscript fully available?

Reviewer #1: Yes

4. Is the manuscript presented in an intelligible fashion and written in standard English?

Reviewer #1: Yes

5. Review Comments to the Author

Reviewer #1: This paper considers the improvement of DG carrying capacity by distributed flexible resources and proposes an accurate assessment method for the maximum carrying capacity of a DG. The effectiveness of the method is verified through simulation. The problem addressed is interesting, however, this paper needs some necessary revisions, and the comments are as follows:

1、It is necessary to read the full text carefully and check carefully to avoid low-level grammatical errors and improper words and phrases. For example, there is a problem with incomplete display of image labels in line 393.

2、In this paper, the linear simplified evaluation model of maximum carrying capacity is developed as a point of contribution, what is the difference between him and other assessment models in the literature, please add the explanation.

3、In the subsection ''Improved comprehensive sensitivity index'', the rationale for the improved comprehensive sensitivity index is not clear enough, and it is recommended that further additional clarification be provided.

4、How can the validity of the DG maximum load carrying capacity simulation results be determined, and is the methodology used in this paper different from the results obtained by other literature methods?

5、In the subsection ''Comparison of different sensitivity indexes'', the traditional sensitivity indexes are briefly described to give the reader a basic understanding.

6. PLOS authors have the option to publish the peer review history of their article (what does this mean?). If published, this will include your full peer review and any attached files.

Reviewer #1: **Yes: **Leijiao Ge

---

## [Author Response · Author response to Decision Letter 0]

13 Apr 2024

Thank you for your positive comments and valuable suggestions to improve the quality of our manuscript. As per your request, we have attached our relevant data(Relevant data.xlsx) and tex file(other.tex).

Based on your valuable input, we have extensively revised our manuscript. Modifications have been highlighted in yellow in the "revised manuscript with track changes". Please refer to the detailed point-by-point responses in "Response to Reviewers.pdf".

1、It is necessary to read the full text carefully and check carefully to avoid low-level grammatical errors and improper words and phrases. For example, there is a problem with incomplete display of image labels in line 393.

Thanks for your suggestion. We have check the full text carefully and tried our best to polish the language in the revised manuscript. These changes will not influence the content and framework of the paper, so we did not list the changes. In addition, the incomplete display of image labels has been corrected. (page 14, line 423)

2、In this paper, the linear simplified evaluation model of maximum carrying capacity is developed as a point of contribution, what is the difference between him and other assessment models in the literature, please add the explanation.

Thank you for your suggestion. We would like to provide a detailed explanation of the issues concerning the linear simplification evaluation model. The maximum DG hosting capacity assessment method proposed in this paper consists of two parts: firstly, the linear optimization model that can quickly calculate the preliminary maximum DG hosting capacity. This model is based on optimal power flow, which is the same as other assessment models in the literature; secondly, the correction of preliminary assessment results based on recursive method, which allows us to obtain more accurate evaluation results. So the contribution of our paper is to propose a integrated assessment method that combines linear optimization method and recursive method as shown in Figure 1. We apologize for not expressing it clearly.

The main modifications are as follows (page 2, line 57-66). Other modifications have been highlighted in yellow in the "revised manuscript with track changes".

The contributions of this paper are summarized as follows:

(1)An integrated assessment method for maximum DG hosting capacity is proposed that combines the linear optimization method and the recursive method. It can realize fast and accurate calculation of maximum DG hosting capacity in the distribution network.

(2)An improved comprehensive sensitivity index is proposed, which can accurately find the most suitable DG connection node for power correction during the recursive process.

(3) A safety constraint verification method based on safety region is proposed, which can realize rapid safety verification calculations and improve the calculation efficiency of recursive algorithm.

3、In the subsection ''Improved comprehensive sensitivity index'', the rationale for the improved comprehensive sensitivity index is not clear enough, and it is recommended that further additional clarification be provided.

Thank you for your suggestion. We have supplemented the rationale for the improved comprehensive sensitivity index. Please refer to the modifications on pages 8-9, lines 232-248.

4、How can the validity of the DG maximum load carrying capacity simulation results be determined, and is the methodology used in this paper different from the results obtained by other literature methods?

(1)The validity of the simulation results for the maximum DG hosting capacity is determined by the following two aspects:

A. Safety verification of simulation results

Adjust the DG output power according to the simulation results presented in this paper. The state variables of the distribution network are shown in "Response to Reviewers.pdf". It can be observed from Fig 2 and Fig 3 in "Response to Reviewers.pdf" that the distribution network state variables are all within the safety constraints, proving the safety of the simulation results.

B. Accuracy verification of the simulation results

Comparison is made between the integrated assessment method and the linear optimization method commonly used in other literature. The DG output power is adjusted according to the simulation results of the two methods, respectively, and the state variables of the distribution network are calculated. DV and DS are then used to evaluate the distance between the operating state and the safety boundary of the distribution network. The smaller the values of DV and DS, the closer the operating state of the distribution network is to the safety boundary, indicating higher accuracy in the assessment results.

It can be observed from Fig 4 to Fig 7 in "Response to Reviewers.pdf" that the simulation results of the proposed method in this paper have lower values of DV and DS, indicating higher accuracy of the assessment method proposed in this paper. This also validates the effectiveness of the DG maximum load hosting capacity simulation results. 

(2)The difference between the methodology used in this paper and the results obtained by other literature method

The literature [7] adopts an iterative calculation method with increasing customer penetration, conducting DG hosting analysis by gradually raising the customer penetration rate. However, this method did not consider the operational characteristics of each node. The literature [8] applied Monte Carlo simulation method to evaluate the maximum DG hosting capacity of distribution networks, but the accuracy of the calculation results is affected by the upper limit of customer penetration. The literature [22] adopts a linear optimization method. The comparison of the assessment results between the method proposed in this paper and other literature methods is shown in Fig 8 and Fig 9 in "Response to Reviewers.pdf". The calculation results of other literature methods may be either conservative or aggressive. The method proposed in this paper can reduce the conservatism of the calculation results by adjusting the magnitude of the DG correction power ΔPDG in the recursive process. The safety constraint verification during the recursive process ensures that the calculation results do not become too aggressive and violate safety constraints.

5、In the subsection ''Comparison of different sensitivity indexes'', the traditional sensitivity indexes are briefly described to give the reader a basic understanding. 

Thank you for your suggestion. We have supplemented the introduction of the traditional sensitivity index in the subsection "Comparison of different sensitivity indexes". Please refer to the modifications on pages 13-14, line 406-413.

---

## [Decision Letter · Decision Letter 1]

9 Sep 2024

PONE-D-24-01853R1Research on Assessment Method of Maximum Distributed Generation Hosting Capacity in Distribution System With High Shares of Renewables and Power ElectronicsPLOS ONE

Dear Dr. He,

Thank you for submitting your manuscript to PLOS ONE. After careful consideration, we feel that it has merit but does not fully meet PLOS ONE’s publication criteria as it currently stands. Therefore, we invite you to submit a revised version of the manuscript that addresses the points raised during the review process.

**ACADEMIC EDITOR: **

Please address the comments from reviewersThe quality of this manuscript should be improved in English and presentations, too.  

We look forward to receiving your revised manuscript.

Kind regards,

Jianguo Wang, PhD

Academic Editor

PLOS ONE

Journal Requirements:

Reviewers' comments:

Reviewer's Responses to Questions

**Comments to the Author**

1. If the authors have adequately addressed your comments raised in a previous round of review and you feel that this manuscript is now acceptable for publication, you may indicate that here to bypass the “Comments to the Author” section, enter your conflict of interest statement in the “Confidential to Editor” section, and submit your "Accept" recommendation.

Reviewer #2: (No Response)

Reviewer #3: All comments have been addressed

2. Is the manuscript technically sound, and do the data support the conclusions?

Reviewer #2: (No Response)

Reviewer #3: Yes

3. Has the statistical analysis been performed appropriately and rigorously? 

Reviewer #2: (No Response)

Reviewer #3: Yes

4. Have the authors made all data underlying the findings in their manuscript fully available?

Reviewer #2: (No Response)

Reviewer #3: Yes

5. Is the manuscript presented in an intelligible fashion and written in standard English?

Reviewer #2: (No Response)

Reviewer #3: Yes

6. Review Comments to the Author

Reviewer #2: The paper has been substantially improved after the first revision. A few minor comments are listed below:

1. Do the 'safety constraints' and 'safety region' in Fig. 3 refer to the same thing? If they are two different things, this needs to be made clear.

2. Adding a few sentences on the organization of the paper would be helpful.

3. It would be better if the manuscript could provide direction on future research or the next steps following this study's completion in the conclusion. Please consider providing some guidance in this area.

Reviewer #3: I'm very glad to review the paper in greater depth because the subject is interesting, and the submission is worthy of publication. The following are my suggestions:

1. Some discussion of the right-hand side of Eq(2) would help the reader understand the equation.

2. Please consider citing references to simplify the derivation process of Eq(17)–Eq(19).

3. The reviewer notices that the parameters Vlim and Slim are used multiple times throughout the paper. In which scenarios do these parameters represent upper bounds, and in which scenarios do they represent lower bounds?

7. PLOS authors have the option to publish the peer review history of their article (what does this mean?). If published, this will include your full peer review and any attached files.

Reviewer #2: No

Reviewer #3: No

---

## [Author Response · Author response to Decision Letter 1]

14 Oct 2024

Dear Editors and Reviewers,

Thank you for your valuable comments and suggestions on our manuscript. We appreciate the time and effort you have spent in reviewing our work. Based on your valuable input, we have extensively revised our manuscript. Modifications have been highlighted in yellow in the "revised manuscript with track changes". The detailed point-by-point responses are listed below:

Reply to Reviewer #2

Comment 1:

Do the 'safety constraints' and 'safety region' in Fig. 3 refer to the same thing? If they are two different things, this needs to be made clear.

Response 1:

Thank for your suggestion. According to the definition in Eq(2), the upper bound ai,max and the lower bound ai,min of the distribution network state variable for both the 'safety constraints' and 'safety region' are derived from Eq(7) and Eq(8). However, the gi(Wt) for the 'safety constraints' is obtained through power flow calculation, while the gi(Wt) for the 'safety region' is calculated using the safety domain model. (page 9, line 280-282)

Comment 2: 

Adding a few sentences on the organization of the paper would be helpful.

Response 2:

Thank you for your suggestion. We have added a description of the organization of the paper. (page 2-3, line 66-71)

The remainder of this paper is organized as follows: Section 2 introduces the concepts of the distribution network operating point and the maximum DG hosting operating point. Section 3 presents the model for assessing maximum DG hosting capacity. Section 4 presents the proposed methodology, describing a two-step method that combines the linearization method and the recursive method. Section 5 presents the test system, numerical results, and discussion.

Comment 3: 

It would be better if the manuscript could provide direction on future research or the next steps following this study's completion in the conclusion. Please consider providing some guidance in this area.

Response 3:

Thank you for your suggestion. We have added an introduction about the direction on future research. (page 14, line 439-441)

To accurately assess the maximum DG hosting capacity in the distribution network, this paper proposes an two-step method that combines the linearization method and the recursive method. Firstly, the maximum DG hosting capacity in the distribution network is preliminarily calculated using linear optimization method, and then the preliminary calculation results are precisely corrected using recursive method. During the recursive process, an improved comprehensive sensitivity index is proposed to find the most suitable DG connection node for power correction. The improved comprehensive sensitivity index can accurately reflect the impact of different DG connection nodes on the distribution network state variables and achieve precise identification of weak points. Additionally, this paper proposes a safety constraint verification method based on security regions, which enhances the calculation speed of the recursive algorithm. Finally, the proposed method was tested and validated on the IEEE 33-bus distribution network. The simulation results indicate that the proposed method can accurately assess the maximum DG hosting capacity in the distribution network. The improved comprehensive sensitivity exhibits significant performance improvement over traditional sensitivity index in enhancing the accuracy of calculating the maximum DG hosting capacity. To further enhance the robustness of the assessment results, the next step of this study will focus on researching an evaluation method for DG hosting capacity in distribution networks that considers various uncertainties.

Reply to Reviewer #3

Comment 1: 

Some discussion of the right-hand side of Eq(2) would help the reader understand the equation.

Response 1:

Thank you for your suggestion. We have added an introduction to the right-hand side of Eq (2). The revisions are as follows: (page 3, line 89-91)

where ΩS is the distribution network safety region, gi(Wt) represents the value of the distribution network state variable ai when the operating point is Wt, ai,max and ai,min are the upper and lower limits of the distribution network state variable ai, respectively.

Comment 2: 

Please consider citing references to simplify the derivation process of Eq(17)–Eq(19).

Response 2:

Thank you for your suggestion. We have simplified the derivation process of Eq (17) - Eq.(19) as per your recommendation. (page 6, line 178-179)

Comment 3: 

The reviewer notices that the parameters Vlim and Slim are used multiple times throughout the paper. In which scenarios do these parameters represent upper bounds, and in which scenarios do they represent lower bounds?

Response 3:

Thank you for your suggestion. The theoretical analysis in the this paper indicates that an increase in DG output power will lead to a rise in node voltage and an increase in reverse power on the feeder. Therefore, for the assessment of DG hosting capacity, Vlim should be set as the upper limit of voltage, and Slim should be set as the upper limit of reverse power on the line. When the assessment target is the hosting capacity for distribution network loads or electric vehicle integration, Vlim should be set as the lower limit of voltage, while Slim should be set as the upper limit of forward power on the line.

Changes to the reference list

In the reference list, the format of references [4] and [12] has been modified. Reference [23] was a paper from Hunan Electric Power, but it could not be found on Google Scholar, so it has been replaced with relevant current references.

Thank you again for your positive comments and valuable suggestions to improve the quality of our manuscript.

---

## [Decision Letter · Decision Letter 2]

21 Oct 2024

Research on Assessment Method of Maximum Distributed Generation Hosting Capacity in Distribution System With High Shares of Renewables and Power Electronics

PONE-D-24-01853R2

Dear Dr. He,

We’re pleased to inform you that your manuscript has been judged scientifically suitable for publication and will be formally accepted for publication once it meets all outstanding technical requirements.

Kind regards,

Jianguo Wang, PhD

Academic Editor

PLOS ONE

Additional Editor Comments (optional):

Reviewers' comments:

Reviewer's Responses to Questions

**Comments to the Author**

1. If the authors have adequately addressed your comments raised in a previous round of review and you feel that this manuscript is now acceptable for publication, you may indicate that here to bypass the “Comments to the Author” section, enter your conflict of interest statement in the “Confidential to Editor” section, and submit your "Accept" recommendation.

Reviewer #2: All comments have been addressed

Reviewer #3: (No Response)

2. Is the manuscript technically sound, and do the data support the conclusions?

Reviewer #2: Yes

Reviewer #3: Yes

3. Has the statistical analysis been performed appropriately and rigorously? 

Reviewer #2: Yes

Reviewer #3: Yes

4. Have the authors made all data underlying the findings in their manuscript fully available?

Reviewer #2: Yes

Reviewer #3: Yes

5. Is the manuscript presented in an intelligible fashion and written in standard English?

Reviewer #2: Yes

Reviewer #3: Yes

6. Review Comments to the Author

Reviewer #2: The author has provided a detailed response to the reviewer's information，and detailed modifications were made to the main text. Thank you！

Reviewer #3: My doubts have been fully resolved. Thank you for the author's response， and I have no other questions.

7. PLOS authors have the option to publish the peer review history of their article (what does this mean?). If published, this will include your full peer review and any attached files.

Reviewer #2: No

Reviewer #3: No

---

## [Editor Report · Acceptance letter]

24 Oct 2024

PONE-D-24-01853R2 

PLOS ONE

Dear Dr. He, 

I'm pleased to inform you that your manuscript has been deemed suitable for publication in PLOS ONE. Congratulations! Your manuscript is now being handed over to our production team.

Kind regards, 

on behalf of

Dr. Jianguo Wang 

Academic Editor

PLOS ONE